# Motion-R³: Fast and Accurate Motion Annotation via Representation-based Representativeness Ranking

## Abstract

As motion capture data collection becomes more accessible, efficient motion annotation tools are increasingly needed to streamline dataset labeling. In this paper, we propose a data-centric motion annotation method that leverages the inherent representativeness of motion data. Specifically, we introduce Representation-based Representativeness Ranking ($R^3$), which ranks motion samples based on their significance in a learned representation space. To enhance this space, we develop a dual-level motion contrastive learning approach, improving the informativeness of the learned representations. Our method is designed for high efficiency and adaptability, making it particularly responsive to frequent requirement changes. By employing unsupervised contrastive pre-training, we reduce the labeling time for expert oracles while using a lightweight classifier to accelerate annotation predictions. Additionally, we incorporate active learning to recommend more representative data, minimizing the number of required expert annotations while maintaining annotation quality. Experimental results on the HDM05 dataset demonstrate that our approach outperforms state-of-the-art methods in both accuracy and efficiency, enabling agile development of motion annotation models.

## 1 Introduction

Along with the recent AI boom, data-driven character animation has been revolutionized and dominated by deep learning (25; 24; 69; 50; 52; 51). Despite its success, deep learning is known to be data-hungry, which poses challenges for both academia and industry, as high-quality annotated data are usually expensive and difficult to obtain. This is even more challenging for mocap (motion capture) data due to the large amount of data frames obtained from dense captures and the complex annotation procedure where multiple labels could be assigned to a single frame (i.e., an actor may wave while walking). To minimize labor costs in annotation tasks, the best-performing methods resort to machine learning solutions. For example, Müller et al. (37) proposed to use motion templates and dynamic time warping (DTW) distance to segment and annotate motion data; Carrara et al. (8) proposed to use long short-term memory (LSTM) network to predict motion labels. Despite their differences, all these methods are *model-centric* and trained with *expert-picked* data that are not normally accessible in practice.

In this paper, we follow the *data-centric AI philosophy* advocated by Andrew Ng (39) and argue that the performance of motion annotation models can be significantly improved by simply using more representative samples in the training. This could aid annotators in labeling only the most critical motions and thereby reduce the overall cost, which is particularly important with the emergence of large models. Specifically, inspired by the classic farthest point sampling strategy and cluster representatives, we propose a Representation-based Representativeness Ranking ($R^3$) method that ranks all motion data in a given dataset according to their "representativeness" in a learned motion representation space $\mathcal{R}$. To learn a more informative $\mathcal{R}$, we propose a novel dual-level contrastive learning method applied at both the motion sequence level and the frame level. In addition, the motion representation space $\mathcal{R}$ learned by our method is independent of specific motion annotation tasks. This generalizability suggests innate adaptability to a wide range of motion annotation objectives, making it more responsive to frequent requirement changes and enabling agile development of motion annotation models.

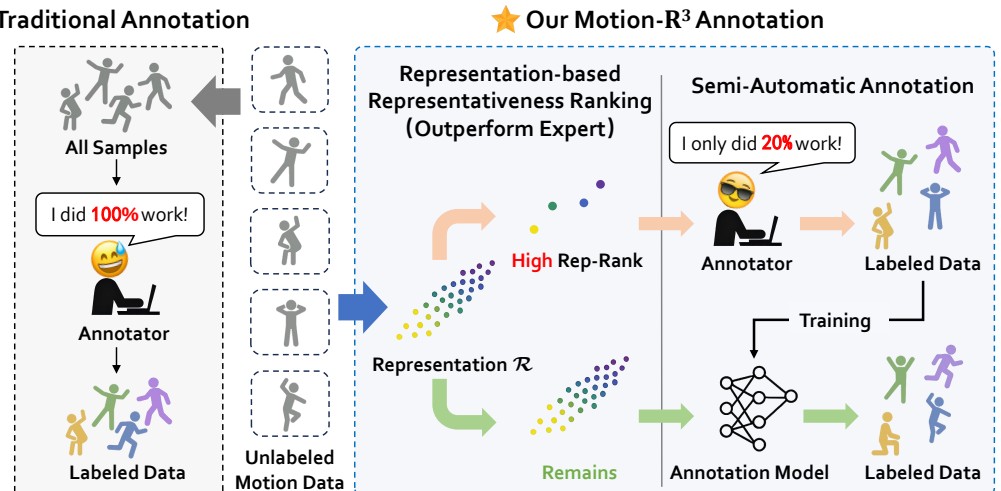

Figure 1: Overview of Motion-R$^3$. Our method selects a small number of highly representative data through a **representation-based representative ranking** then significantly improves the efficiency of action annotation via a coordinated workflow between annotators and annotation models. *Our method achieves over 0.85 micro-F1 score on the HDM05-15 dataset using only 21.57% selected high representative data for training.

Our Motion-R$^3$ method is the first to achieve fast and accurate motion annotation without expert guidance.

Extensive experimental results on the HDM05 and BABEL datasets demonstrate the superiority of our method against the state-of-the-art. In summary, our main contributions include:

- We propose Motion-R$^3$, the first approach to achieve fast and accurate motion annotation without expert guidance, thereby significantly reducing labor costs.

- We propose a novel dual-level motion contrastive learning method that automatically learns a representation space for motion data that is more informative than expert knowledge. This representation space is independent of specific motion annotation tasks, thus demonstrating generalizability across frequent requirement changes.

- We propose a novel representativeness ranking method based on density and distance heuristics, featuring a robust ranking initialization scheme that enables consistently near-optimal annotation accuracy, right from the selection of the very first sample (*i.e.*, seed).

## 2 RELATED WORK

### 2.1 MOTION ANNOTATION

Motion annotation aims to annotate raw and unsegmented motion data with action labels, which is a complex and tedious task as multiple action labels can be assigned to the same piece of data (5; 73). To address its challenges, a straightforward idea is to first divide the raw mocap data into action segments and then classify them respectively. For example, the sliding window method was employed to divide raw mocap data into overlapping action segments (37; 35; 62; 65) or non-overlapping semantic segments (41; 6; 15). The classification of segmented action segments is usually referred to as an action recognition task, which aims to classify each action segment to the correct action category across different spatio-temporal configurations (*e.g.,* velocity, temporal or spatial location) (9; 46; 74; 17; 18; 28; 4; 34; 40; 49). Compared to traditional model-based methods (64; 59; 28) and classifiers (45; 68; 57), state-of-the-art action recognition methods resort to deep convolutional neural networks (29; 46) and LSTM neural networks (74; 40; 48) to effectively model spatial and temporal motion features, as deep learning has demonstrated its power in identifying complex patterns in multimedia data (2; 3). On the other hand, frame-based motion annotation methods have recently

gained popularity as they are more fine-grained and can predict the probabilities of each action per frame directly. The classification tasks in these methods are usually implemented by vector machines (47), linear classifiers (72), structured streaming skeletons (71), LSTM networks (32; 49; 8), etc. Furthermore, flow-based methods can identify motion before the motor behavior ends (31) and even predict future action (27; 61).

In this work, we investigate an important but under-explored problem in motion annotation, *i.e.,* the representativeness of mocap data points. We demonstrate that the performance of motion annotation can be significantly improved by simply picking more representative samples for training.

## 2.2 CONTRASTIVE LEARNING

Contrastive learning is an unsupervised representation learning method that can learn high-quality feature spaces from unlabeled data (54; 55; 63; 23; 10; 12). Contrastive Learning has made great progress in the field of computer vision (58; 70; 22; 63; 21; 54; 7; 10; 23; 12; 11; 13). And Momentum Contrastive Paradigm (MoCo) (23; 12) facilitates contrastive unsupervised learning through a queue-based dictionary lookup mechanism and momentum-based updates.

Contrastive learning has already been applied and achieved promising results in motion-related tasks. MS2L (33) integrates contrastive learning into a multi-task learning framework; AS-CAL (44) uses different backbone sequence augmentations to generate positive and negative pairs; Thoker et al. (53) perform representation learning in a graph-based and sequence-based mode using two different network architectures in a cross-contrasted manner. Recently, SkeletonCLR (30) learns skeleton sequence representations through a momentum contrast framework. In a concurrent work, AimCLR (20) extends SkeletonCLR with an energy-based attention-guided casting module and nearest neighbor mining. BYOL (36) extends representation learning for skeleton sequence data and proposes a new data augmentation strategy, including two asymmetric transformation pipelines.

In this work, we propose a novel *dual-level* motion contrastive learning approach which extends MoCo (23; 12) to motion data and implements contrastive learning at both sequence and frame levels, which works as the basis of the proposed Motion-$R^3$ method.

## 2.3 MOTION REPRESENTATION

Motion representation can be used in many applications for indexing, temporal segmentation, retrieval, and synthesis of motion clips, using methods such as weighted PCA (19) and comparative learning (1). Bernard et al. (5) operates a combination of hierarchical algorithms to create search groups and extract motion sequences. Zhou et al. (73) applies alignment clustering analysis to action segmentation and expands standard kernel k-means clustering through dynamic time warping (DTW) kernel to achieve temporary variance. Holden et al. (26) utilizes an automatic convolutional encoder to learn a variety of human motion manifolds as a motion priori to resolve ambiguity. Choi and Kwon (14) converts raw motion data into animated short films called motion clips, which involve temporal segmentation, emphasizing the main motion, minimizing data size, and utilizing a ranking algorithm for query retrieval. Won et al. (60) represents motion as a high-level scene specification by manually constructing or automatically extracting it, which consists of sentence-like structures involving verbs, subjects, and objects, in order to recommend a small and diverse selection of the highest quality scenes.

# 3 OUR MOTION-$R^3$ METHOD

Let $D = (x_1, x_2, ..., x_N)$ be a motion dataset consisting of $N$ motion sequences, $x_i = (s_{i,1}, s_{i,2}, ..., s_{i,T})$ be a motion sequence consisting of $T$ consecutive skeleton pose frames, $s_{i,j} \in \mathbb{R}^{J \times 3}$ $(j = 1, 2, ..., T)$ be the 3D coordinates of the $J$ body joints of a skeleton pose, we aim to assign a binary label vector $\mathbf{c_{i,j}} = \{0,1\}^m$ to each skeleton pose $s_{i,j}$ where $c_{i,j,k} = 1$ $(k = 1, 2, ..., m)$ if $s_{i,j}$ belongs to the $k$-th class of $m$ pre-defined motion types. To minimize labour costs, we assume only a small portion of $D$ are manually annotated as $D_{\text{train}}$ and the rest can be automatically annotated by a machine learning model trained with $D_{\text{train}}$ as the training set.

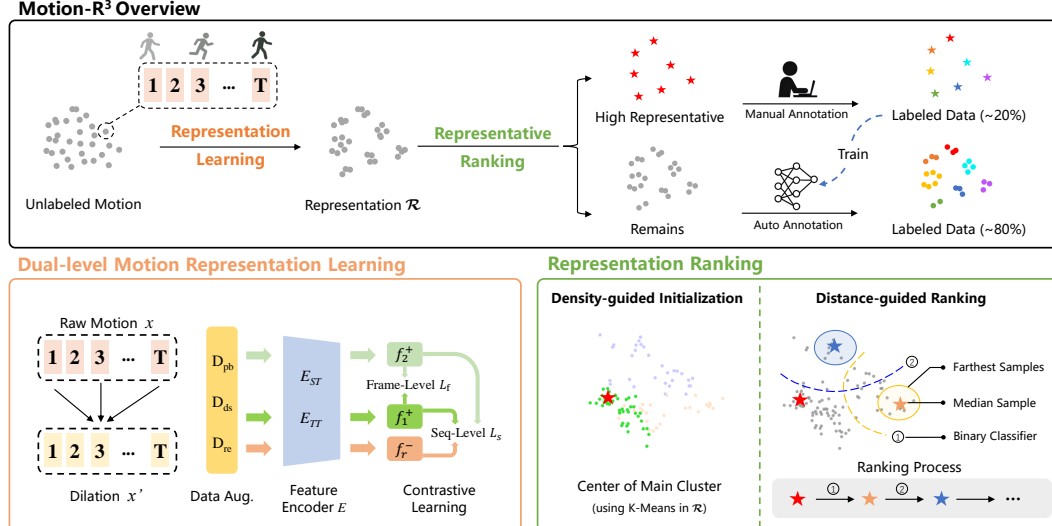

Figure 2: Overview of our Representation-based Representativeness Ranking ($R^3$) method. 1) We first utilize Dual-level Motion Representation Learning to provides effective motion representation $\mathcal{R}$, then 2) apply Representation Ranking on $\mathcal{R}$ to pick few amount of high representativeness samples for manual annotation. Finally, the annotation model trained with manual annotated data will finish the remains work.

## 3.1 OVERVIEW

Our method aims to learn a representativeness ranking of motion sequences $x_i \in D$ in an unsupervised manner, for the task of fast and accurate motion annotation (Fig. 2). Unlike previous methods (8) which select $D_{\text{train}}$ by expert visual inspection, we argue that picking the more *representative* ones for manual annotation not only reduces the labour and time costs but also increases the model's accuracy. Alg. 1 shows the pseudo-code of our $R^3$ method.

**Representation Learning**    We first train a feature encoder $E$ which learns a representation space $\mathcal{R}$ for $x_i$ in an unsupervised manner. For the learning of feature encoder $E$ and motion representation space $\mathcal{R}$, we adopt one of the latest contrastive learning approach: Momentum Contrast (MoCo) (23; 12), which has recently demonstrated superior performance and generalization abilities in computer vision tasks. Nevertheless, MoCo was designed for computer vision tasks and only works on the 2D grid-like image data. Thus, it is non-trivial to acclimatize it to motion sequences. Addressing this issue, we propose a *dual-level* motion contrastive learning approach which extends MoCo to motion data and implements contrastive learning at both sequence and frame levels, which is depicted in the next subsection.

**Representativeness Ranking**    We next map motion sequence to $\mathcal{R}$. Inspired by the classic farthest point sampling strategy, we implement our $R^3$ method by progressively including $x_i \in D$ to a ranked motion dataset $\hat{D}$, where $x_i$ is the median of a small proportion (*i.e.,* 10%) of "farthest" (*i.e.,* the most representative) motion sequences to the ranked ones in $\hat{D}$. The ranking is then determined by the order in which $x_i$ is included into $\hat{D}$. Note that we search the "farthest" points via binary classification to avoid the high computational costs of traditional farthest point sampling methods that consume $O(n^2)$ time and decrease it to $O(n)$. Additionally, using the median instead of directly using the "farthest" data enables our method to more effectively sample from high-density regions rather than selecting outliers.

**Motion Annotation with $R^3$**    For motion annotation, we first assign motion sequences to human annotators according to the ranking $\hat{D}$ and get $\hat{D}_{\text{train}}$. Then, we train a low-cost and simple classifier $C_{\text{simple}}$ using the learned representation $\mathcal{R}$ and $\hat{D}_{\text{train}}$ to annotate the remaining motion sequences automatically.

## 3.2 DUAL-LEVEL MOTION REPRESENTATION LEARNING

In a nutshell, contrastive learning assumes that a good data representation has two properties: similar data points should be close to each other in the feature space, while different data points should be far from each other. Accordingly, it proposes to fulfil the two properties by minimizing the distances among positively augmented samples and maximizing those among negatively augmented samples. Building on this idea, MoCo (23; 12) shows that the performance of contrastive learning can be boosted by maintaining a large and consistent dictionary of negatively augmented samples, which is implemented by the incorporation of a queue and a momentum encoder.

Thus, the extension of MoCo to *motion data* boils down to three questions: i) how to select a proper backbone network for feature encoding? ii) how to design the positive and negative data augmentation methods? iii) how to measure the distances between samples in the feature spaces (*i.e.,* the contrastive loss)? Fig. 2 (1) shows a data flow diagram of our solution, which is detailed as follows.

### 3.2.1 DILATED (MOMENTUM) FEATURE ENCODER.

Since motion data are usually captured at a high sampling rate (*e.g.,* 120 FPS), the differences between adjacent frames are tiny, which causes ambiguities that confuse the model in identifying the action of a single frame. To clarify such ambiguity, we borrow the idea of *dilated* convolution (66) and enhance each input frame with its context information (*i.e.,* dilated joint trajectory) in a time window $t$ centered at the current frame. Specifically, assuming the sampling rate is $r = 120$ FPS, we employ a dilation factor $l$ that enhances input frame $s_{i,j}$ with its context information as

$$s'_{i,j} = (s_{i,j} - s_{i,j-nl}, ..., s_{i,j} - s_{i,j-l}, s_{i,j}, s_{i,j+l} - s_{i,j}, ..., s_{i,j+nl} - s_{i,j}) \tag{1}$$

where $n = \lfloor t \cdot r/l \rfloor$, $\lfloor \cdot \rfloor$ is a flooring function, $\pm(s_{i,j} - s_{i,j+kl})$ denotes the dilated joint trajectory, $k = \{-n, -n+1, ..., 0, ..., n\}$. We use $x'_i = (s'_{i,1}, s'_{i,2}, ..., s'_{i,T})$ as the input to our (momentum) feature encoders. We replace the Vision Transformer (16) with a similar method as Spatial Temporal Transformer (42) as our feature encoder, for its success in modeling the dependencies among skeleton joints. Specifically, after being embedded in a two-layer MLP network, its models the relationships among joints of a single skeleton in each frame with the so-called Spatial Transformer ($E_{ST}$) module and those among the same joints across different frames in $x'_i$ with its Temporal Self-Transformer ($E_{TT}$) module.

Since our motion data is a motion *sequence* consisting of consecutive *frames* of skeleton poses, we propose to implement contrastive learning at both levels as follows.

### 3.2.2 SEQUENCE-LEVEL CONTRASTIVE LEARNING

We show the data augmentation methods and loss design of the proposed sequence-level contrastive learning method below.

**Sequence-level Data Augmentation.** Similar to those in computer vision tasks (23; 10; 12), the key challenge of motion data augmentation is to disentangle the inherent patterns of a skeleton from its different *sequences*, *i.e.,* the different appearances of the same pattern. To create such different sequences:

i) We propose a *perturbation* data augmentation strategy with the rationale that motion semantics are robust against small perturbations. Specifically, we apply two stochastic perturbations, data missing and disorder, to each input frame $s'_i$ according to $p_i \sim \mathcal{U}[0, 1]$ as:

$$pb(s'_i, p_i) = \left\{ \begin{array}{ll} 0, & p_i < t_{pb} \cdot t_{md} \\ s'_j, & t_{pb} \cdot t_{md} \leq p_i < t_{pb} \end{array} \right. , j \sim \mathcal{U}\{1, T\} \tag{2}$$

where $t_{pb} = 0.15 \in [0, 1]$ is the probability threshold that $s'_i$ is perturbed, $t_{md} = 0.9 \in [0, 1]$ is the probability threshold that missing data perturbation is applied, $t_{pb} \cdot t_{md}$ means that $s'_i$ is perturbed with missing data perturbation, the replacement of $s'_i$ with $s'_j$ denotes the disorder perturbation, $\mathcal{U}\{1, T\}$ denotes a discrete Uniform distribution from 1 to $T$. Let $p = \{p_i\}_{i=1}^T$, we have

$$\mathcal{D}_{pb}(x', p) = (pb(s'_1, p_1), pb(s'_2, p_2), ..., pb(s'_T, p_T)) \tag{3}$$

ii) Inspired by the fact that human beings can successfully recognize motions at different playback speeds (*i.e.,* the motion semantics are largely independent of the playback speeds), we propose a novel *downsampling* augmentation technique that creates novel sequences of motion data by downsampling them at random rates and offsets:

$$\mathcal{D}_{ds}(x', a, \delta) = (s'_a, s'_{a+\delta}, s'_{a+2\delta}, ..., s'_{a+(n_{ds}-1)\delta}) \tag{4}$$

where $a$ denotes the offset, $\delta$ denotes the downsampling interval, $n_{ds} = 512$ denotes the number of resulting samples. Note that $a + (n_{ds} - 1)\delta \leq T$.

iii) We also propose the *reverse* augmentation that works as a negative augmentation method:

$$\mathcal{D}_{re}(x') = (s'_T, s'_{T-1}, ..., s'_1) \tag{5}$$

With the aforementioned data augmentation methods, we generate two positively augmented sequences $v_1^+, v_2^+$ and a negatively augmented sequence $v_r^-$ as: $v_1^+ = \mathcal{D}_{pb}(\mathcal{D}_{ds}(x', a^1, \delta^1), p^1), v_2^+ = \mathcal{D}_{pb}(\mathcal{D}_{ds}(x', a^2, \delta^2), p^2), v_r^- = \mathcal{D}_{re}(\mathcal{D}_{pb}(\mathcal{D}_{ds}(x', a^3, \delta^3), p^3))$, where $a^i, \delta^i, p^i$ denote different parameters generated randomly.

We encode these augmented sequences and get their normalized features with dilated (momentum) feature encoder $E$:

$$f_1^+ = \frac{E(v_1^+)}{\|E(v_1^+)\|}, f_2^+ = \frac{E(v_2^+)}{\|E(v_2^+)\|}, f_r^- = \frac{E(v_r^-)}{\|E(v_r^-)\|}. \tag{6}$$

**Sequence-level Contrastive Loss.** We design our loss function based on an InfoNCE loss:

$$\mathcal{L}_s = -\log \frac{\exp(f_1^+ \cdot f_2^+/\tau)}{\exp(f_1^+ \cdot f_2^+/\tau) + \sum_{i=1}^{K} \exp(f_1^+ \cdot f_i^-/\tau)} \tag{7}$$

where $\cdot$ denotes the measurement of cosine similarity, $\tau$ is a temperature softening hyper-parameter and $i$ denotes the indices of the negative samples $f_i^-$ maintained in the queue $Q$ of size $K$ that $Q = f_r^- \frown \{f_1^-, f_2^-\}$ where $\frown$ denotes the enqueue operation, $\{f_1^-, f_2^-\}$ denotes the positive samples generated previously but are used as negative samples for $f_1^+$ as they are generated from different $x'$.

### 3.2.3 FRAME-LEVEL CONTRASTIVE LEARNING

**Frame-level Data Augmentation.** Leveraging the *local consistency* among consecutive frames in a motion sequence (*i.e.,* the actions in a small neighbourhood share similar motion semantics), for the feature $f_{1,i}^+$ of each frame $s'_i$, we define $f_{2,j}^+$ are its *positive* samples if $j \in \Omega_+$ and $\Omega_+ = \{j | t_{nb} > |i - j|\}$, where $t_{nb} = 12$ is the size of the neighbourhood. It is worth noting that contrastive learning only encourages positive samples to be close in the representation space and does not include negative samples in the loss calculation.

**Frame-level Contrastive Loss.** Accordingly, we design our frame-level local consistency loss as:

$$\mathcal{L}_f = -\log \sum_i \sum_{j \in \Omega_+} \exp\left(f_{1,i}^+ \cdot f_{2,j}^+/\tau\right) \tag{8}$$

Combining the sequence-level loss $\mathcal{L}_s$ and the frame-level loss $\mathcal{L}_f$, the overall training loss is $\mathcal{L} = \mathcal{L}_s + \mathcal{L}_f$.

### 3.3 RANKING WITH DENSITY AND DISTANCE HEURISTICS

Quantifying representativeness directly is challenging as it is an abstract concept. To enable representativeness ranking, we propose to measure representativeness in the informative representation space $\mathcal{R}$ obtained in Sec. 3.2 from two aspects:

- **Density.** Data points located in high-density regions tend to have high representativeness as they can accurately represent the characteristics of their neighbours; and vice versa.
- **Distance.** The most representative samples should have a large inter-class distance, so as to cover the complete data distribution with as few samples as possible.

Leveraging the above two heuristics, we propose an unsupervised representativeness ranking method as follows. The method consists of an initialization step to select a good seed, followed by an iterative ranking process to expand the set of representative data points (Fig. 2). The pseudocode of Representative Ranking is given in the Appendix.

**Density-guided Initialization.** We aim to identify a highly representative sample (*i.e.*, seed) from among the unlabeled motion sequences $D$ to initialize the ranking. As discussed, such samples generally reside in high-density regions (*i.e.*, clusters) when mapped to $R$. Accordingly, we first apply k-means clustering to $D$'s embeddings in $R$, segmenting $D$ into $k$ distinct clusters $D^{(1)}, ..., D^{(k)}$, and then select the largest cardinality (indicating high density) cluster as the primary cluster :$D_m = \arg\max_{D^{(i)}} |D^{(i)}|, i \in \{1, 2, ..., k\}$. Then we choose the sample closest to the center of $D_m$ as the initial sample.

**Distance-guided Ranking with Density-based Regularization.** After initialization, subsequent ranking iterations require distance-based guidance to promote diversity among the representatives, so as to enable a swift coverage of the entire data distribution. However, the density heuristic used in initialization must be retained to preserve local representativeness at the same time. To achieve this, we propose a novel representative ranking method compromising two steps: i) we train a binary classifier $C$ to distinguish between the set of to-be-ranked motion data $D$ (labeled 0) and ranked motion data $\hat{D}$ (labeled 1). In this way, data with a lower predicted value (closer to 0) is considered more representative to $\hat{D}$. ii) We apply density-based regularization by marking a small portion (*i.e.*, 10%) of samples with the lowest predicted values as candidates and selecting their median as the next representative, avoiding picking outliers from low-density areas (see Fig. 4).

## 4 EXPERIMENTS & RESULTS

### 4.1 IMPLEMENTATION DETAILS

We conduct experiments on a PC with an Intel i7-7700 CPU and a Nvidea TESLA P40 GPU. We implement our method with PyTorch. We have included more experimental details in the supplementary materials. Following (8), we evaluate our method on the three variants of the HDM05 dataset (38):

- **HDM05-15**: 102 motion sequences, 15 classes, 68 minutes (491,847 frames);
- **HDM05-65**: 2,345 motion sequences, 65 classes, 156 minutes (1,125,652 frames);
- **HDM05-122**: 2,238 motion sequences, 122 classes, 156 minutes (1,125,652 frames).

Note that HDM05-65 and HDM05-122 contain the same data but have different labels. We use the global joint positions as the raw motion representation. This is because joint position representation is easier to transfer to different skeleton (e.g. world-aligned skeletons and axis-aligned skeletons) than rotation representation.

**Motion Data Preprocessing** We follow (46) to preprocess the motion data, normalizing the skeleton size, root position and orientation. This ensures that joint positions and angle values are effectively equivalent. Thus, we use joint positions in all our experiments.

### 4.2 EFFECTIVENESS OF MOTION REPRESENTATION LEARNING

To justify the effectiveness of our motion representation, we first compare our method with two state-of-the-art motion representation methods, namely Convolutional Autoencoders (CA) (26) and Deep Motifs and Motion Signatures (DMMS) (1), on the HDM05 dataset under the same conditions. As Table 1 shows, i) our method outperforms both baselines and achieves the highest accuracy; ii) DMMS suffers from a significant performance drop when being applied to relatively complex scenarios (HDM05-65 and HDM65-122).

We also justify the necessity of motion representation learning by comparing our method with its naive variant which applies the proposed representativeness ranking algorithm to the raw motion

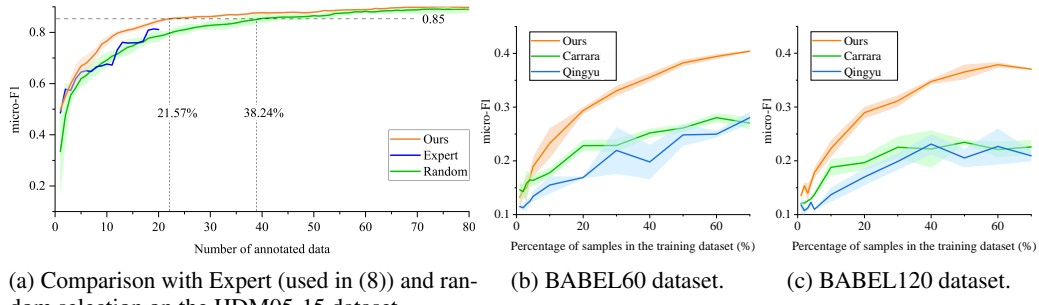

(a) Comparison with Expert (used in (8)) and random selection on the HDM05-15 dataset.

(b) BABEL60 dataset.

(c) BABEL120 dataset.

Figure 3: Comparison with Carrara (8) and Qingyu (67) on the BABEL60 and BABEL120 datasets.

Table 1: Comparison between ours and two state-of-the-art motion representation: CA (26) and DMMS (1). All experiments use the same 20% high-representativeness data for training, and then test the annotation accuracy on the remaining 80% of the data.

| Method | HDM05-15 | HDM05-65 | HDM05-122 |
|--------|----------|----------|-----------|
| CA | 72.23 | 75.08 | 71.88 |
| DMMS | 42.17 | 2.39 | 1.43 |
| Ours | **84.37** | **83.77** | **82.06** |

data directly. Experimental results on the HDM05-15 dataset show that the micro-F1 accuracy on the raw data is 61.62%, which is significantly lower than the 84.37% achieved when using motion representation. This confirms the necessity of motion representation learning in better extracting motion features for the subsequent tasks of ranking and annotation.

## 4.3 MOTION-$R^3$ V.S. STATE-OF-THE-ART MOTION ANNOTATION

As shown in Fig. 3, the proposed representativeness ranking consistently outperforms expert ranking. In addition, our method requires significantly less training data to reach strong annotation accuracy (micro-F1 > 0.85), using only 21.57% of labels versus 38.24% for random selection - nearly halving the annotation effort. This substantial reduction in motion annotation workload signifies the potential of our method in creating larger motion datasets for downstream tasks.

We also evaluate the performance our method and SOTA methods under different annotation requirements. The experimental results in Table 2 highlight the minimum amount of data required by our methods (Ours+Carrara and Ours+$\mathcal{R}$+MLP) to surpass the accuracy of (8) (Expert+Carrara), as well as their performance when using the same amount of training data as (8). Interestingly, Ours+$\mathcal{R}$+MLP outperforms Ours+Carrara on the HDM05-65 and HDM05-122 datasets, further demonstrating the effectiveness of our data-centric approach and the learned motion representation. The perceived "limited" improvement is due to (8) utilizing significantly more data than necessary to reach performance saturation. Even with reduced data, our method still achieves superior performance.

We also evaluate our method on the **BABEL** (43) dataset, please refer to appendix for detailed results.

## 4.4 ABLATION STUDY

### 4.4.1 DESIGN OF DUAL-LEVEL MOTION REPRESENTATION LEARNING

As Table 3 shows, the experimental results justify the effectiveness of the algorithmic designs of our dual-level contrastive learning method.

The results show that applying contrastive learning at both the sequence and frame levels contributes to the final performance. The four designs on the sequence level play a more important role in boosting the performance of our method, compared with the one on the frame level. The results with the native Moco method show that it is not directly applicable to the task of motion annotation.

Table 2: Comparison with state-of-the-art motion annotation methods: Müller et al. (37) and Carrara et al. (8). Ours[1]: the accuracy when using the same amount of training data as "Expert + Carrara"; Ours[2]: the minimum amount of data required to achieve higher accuracy than "Expert + Carrara"; $\mathcal{R}$+MLP: train a simple Multi-layer perceptron using the learned motion representation space $\mathcal{R}$. Müller[*]: we did not test "Ours[1,2] + Müller" as the source code of (37) was not publicly released.

| Method | | HDM05-15 | | HDM05-65 | | HDM05-122 | |
|---|---|---|---|---|---|---|---|
| Sampling | Annotation | Train (%) ↓ | micro-F1 (%)↑ | Train (%) | micro-F1 (%) | Train (%) | micro-F1 (%) |
| Expert | Müller[*] | 28.57 | 75.00 | - | - | - | - |
| Expert | Carrara | 19.61 | 78.78 | 44.12 | 64.82 | 44.12 | 57.66 |
| Ours[1] | Carrara | 19.61 | 80.50 | 44.12 | 67.00 | 44.12 | 60.70 |
| Ours[2] | Carrara | 15.69 | 79.20 | 40.76 | 65.00 | 42.02 | 58.42 |
| Ours[1] | $\mathcal{R}$+MLP | 19.61 | 84.37 | 44.12 | 71.13 | 44.12 | 68.69 |
| Ours[2] | $\mathcal{R}$+MLP | 15.69 | 79.05 | 25.21 | 65.56 | 22.68 | 59.94 |

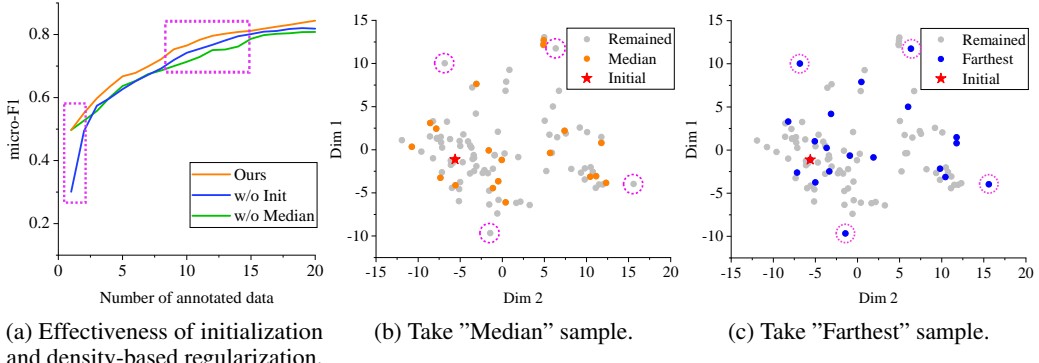

(a) Effectiveness of initialization and density-based regularization.

(b) Take "Median" sample.

(c) Take "Farthest" sample.

Figure 4: Ablation Study on the design of our representativeness ranking method. (b & c) Visualization of top-20 ranking results of the "Median" and "Farthest" ranking strategy on HDM05-15 dataset.

Table 3: Ablation Study on our Dual-level Contrastive Learning. MoCo baseline[*]: naive adaptation of MoCo (23) to our task. The motion data are annotated in the ranked order generated by our strategy.

| Method | micro-F1(%) | macro-F1(%) |
|---|---|---|
| MoCo baseline[*] | 59.31 | 39.54 |
| *+Sequence Level* | 82.07 | 77.09 |
| *+Frame Level* | 84.37 | 77.55 |

### 4.4.2 DESIGN OF REPRESENTATIVENESS RANKING

As shown in Fig. 4a, without initialization, the ranking does not start from highly representative samples, which affects the quality of subsequent ranking; without Density-based Regularization (*i.e.*, "Median" in the figure), even starting from highly representative samples, the absence of density constraints can introduce outliers samples, prevents this advantage from being sustained during the ranking.

## 5 CONCLUSION

In this paper, we propose a novel motion annotation method, namely Motion-R[3], which shows that the performance of motion annotation can be significantly improved by using the more representative training samples extracted by our Representation-based Representativeness Ranking (R[3]) method. Our R[3] method relies on an informative representation space learned by the proposed novel dual-level motion contrastive learning method. Thanks to its high efficiency, our Motion-R[3] method is particularly responsive to frequent requirement changes and enables agile development of motion annotation models, which sheds light on a new working paradigm for both academia and the industry.

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

## 6 ETHICS STATEMENT

All authors of this paper have carefully read and fully adhere to the ICLR Code of Ethics (https://iclr.cc/public/CodeOfEthics).

Notably, the user study described in the appendix has undergone and obtained approval from the ethical review board of the authors' affiliated institution. Prior to participating in the action annotation tasks in the study, all volunteers were provided with comprehensive information about the study's purpose, procedures, potential risks, and benefits. Each volunteer voluntarily signed an informed consent form to confirm their participation. Additionally, all participants received reasonable monetary compensation for their time and efforts in completing the annotation tasks, ensuring fair treatment in line with ethical research practices.

## 7 REPRODUCIBILITY STATEMENT

The authors guarantee that all results presented in this paper are reproducible, and have taken targeted measures to facilitate the replication of the work by other researchers, with details distributed across the main text and appendix. To ensure the reproducibility of the work presented in this paper, the authors will release the code of Dual-Level Motion Representation Learning and Representativeness Ranking on Github upon paper acceptance.

# 8 APPENDIX

## 8.1 USER STUDY

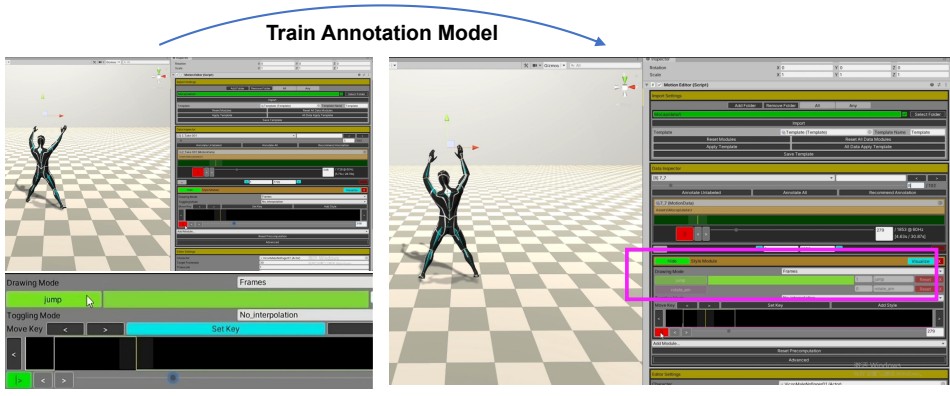

① Annotate 'jump' motion    ② Next jump motion is automatically annotated

Figure 5: Illustration of Our Motion-R$^3$ Driven Annotation Tool. Once trained with high representativeness samples, the annotation model can accurately annotate subsequent samples.

We developed a prototype motion annotation toolkit in Unity3D powered by our Motion-R$^3$ for semi-automated motion annotation. In each round of annotation, three motion sequences will be pushed to the annotator according to Density-guided Initialization Representativeness Ranking. The annotated data were used to update the annotation model, then preliminary annotations were made for subsequent motion sequences, thereby reducing the labor costs.

We conducted annotation experiment on HDM05-15 dataset with our annotation tool. We recruited 10 volunteers with an average age of 23.6. They are all graduate students majoring in software engineering. All volunteers provided informed consent by signing the consent form prior to participation. Volunteers were compensated with a reasonable stipend for their participation. The study protocol underwent and was approved by the institutional ethics review board.

The volunteers were evenly divided into two groups:

- Experimental group: Using Motion-R$^3$.
- Control group: Pushed three random motion sequences to the volunteers each time, no preliminary annotation with annotation model.

Table 4: Average annotation time per sequence for Experimental vs. Control groups

| Group | Average time per sequence (s)reduction |
|---|---|
| Experimental | 103.2 (35.82% ↓) |
| Control | 160.8 |

The results show that the average annotation time per sequence in the experimental group is 103.2 seconds, representing a 35.82% reduction from the control group's 160.8 seconds. Furthermore, **100% of experimental participants reported that fatigue during annotation decreased as the annotation model's accuracy improved**, since they mainly needed to verify annotation model's outputs and correct small number of errors.

## 8.2 EXPERIMENT ON BABEL DATASET

To demonstrate the superiority of our Motion-R$^3$ method, we quantitatively compare it with two state-of-the-art motion annotation methods (67; 8) on both BABEL60 and BABEL120 datasets. We divide train, test and validation dataset according to BABEL. We pretrained our model on the train

Table 5: Results on BABEL (43): We report our Motion-R$^3$ on both BABEL60 and BABEL120.

| | Motion-R$^3$ (Ours) | | Carrara (8) | | Qingyu (67) | |
|---|---|---|---|---|---|---|
| Train (%) | BABEL60 (%) | BABEL120 (%) | BABEL60 (%) | BABEL120 (%) | BABEL60 (%) | BABEL120 (%) |
| 1 | $13.23 \pm 2.33$ | $13.51 \pm 0.34$ | $14.61 \pm 1.66$ | $12.10 \pm 0.35$ | $11.50 \pm 0.53$ | $11.81 \pm 1.76$ |
| 2 | $14.58 \pm 1.26$ | $15.36 \pm 0.71$ | $14.27 \pm 1.38$ | $12.15 \pm 0.50$ | $11.28 \pm 0.52$ | $10.73 \pm 0.68$ |
| 3 | $14.90 \pm 2.30$ | $13.99 \pm 1.84$ | $15.81 \pm 2.47$ | $12.55 \pm 0.01$ | $11.94 \pm 1.08$ | $11.21 \pm 0.48$ |
| 4 | $16.72 \pm 1.15$ | $15.93 \pm 0.74$ | $16.44 \pm 1.41$ | $12.94 \pm 0.41$ | $12.45 \pm 0.36$ | $12.26 \pm 0.73$ |
| 5 | $18.97 \pm 1.83$ | $17.80 \pm 0.98$ | $16.38 \pm 1.06$ | $13.72 \pm 0.54$ | $13.41 \pm 0.99$ | $10.98 \pm 0.06$ |
| 10 | $23.30 \pm 2.79$ | $22.30 \pm 1.57$ | $17.79 \pm 0.94$ | $18.79 \pm 1.55$ | $15.51 \pm 1.60$ | $13.71 \pm 1.35$ |
| 20 | $29.37 \pm 0.60$ | $28.95 \pm 1.09$ | $22.84 \pm 1.04$ | $19.67 \pm 1.38$ | $16.92 \pm 0.21$ | $17.00 \pm 1.58$ |
| 30 | $33.04 \pm 0.95$ | $31.14 \pm 1.05$ | $22.88 \pm 1.20$ | $22.52 \pm 2.01$ | $21.94 \pm 4.36$ | $19.89 \pm 1.56$ |
| 40 | $35.54 \pm 0.80$ | $34.74 \pm 0.47$ | $25.19 \pm 0.86$ | $22.20 \pm 3.46$ | $19.81 \pm 3.19$ | $23.11 \pm 1.91$ |
| 50 | $38.25 \pm 0.60$ | $36.53 \pm 1.37$ | $26.12 \pm 0.72$ | $23.45 \pm 0.47$ | $24.83 \pm 2.03$ | $20.54 \pm 1.71$ |
| 60 | $39.43 \pm 0.45$ | $37.86 \pm 0.50$ | $28.05 \pm 0.84$ | $22.11 \pm 1.43$ | $24.97 \pm 0.63$ | $22.67 \pm 3.36$ |
| 70 | $40.42 \pm 0.17$ | $37.05 \pm 0.12$ | $27.03 \pm 1.12$ | $22.57 \pm 1.38$ | $28.05 \pm 1.12$ | $20.93 \pm 1.01$ |

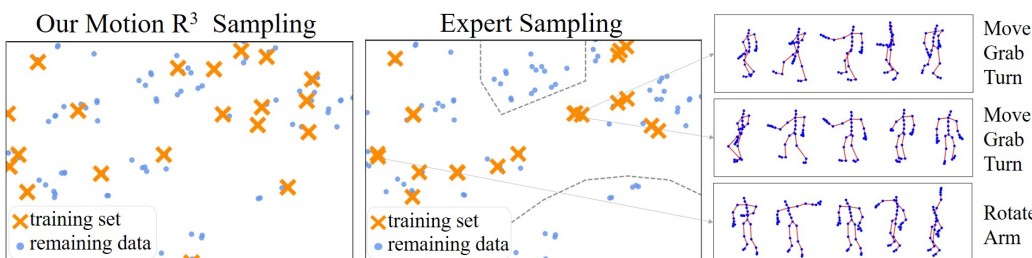

Figure 6: Comparison of sample representativeness of our Motion-R$^3$ method against expert selection (8) in the feature space.

dataset and tested our method on the validation dataset. The results show that our method can achieve 24.74% when using half of the train dataset.

### 8.3 MOTION-$R^3$ V.S. EXPERT RANKING

To qualitatively evaluate the representativeness of samples identified by our Motion-R$^3$ method, we first extracted features from all data in the HDM05-15 dataset to construct the representation space. Subsequently, we employed t-SNE (56) to reduce the dimensionality of the representation space to 2D for visualization purposes. As Fig. 6 shows, the samples selected by our Motion-R$^3$ method are more evenly distributed according to the entire data distribution and are more representative than those selected by the expert (8), which justifies the effectiveness of our Motion-R$^3$ method. In contrast, expert sampling suffers from overlapping samples (*e.g.,* the four sample pairs in the top right corner of Fig. 6-middle). As the skeletons on the right illustrate, these samples share similar motion sequences (Move-Grab-Turn), resulting in redundancy for motion annotation tasks. An important side effect of such redundancy is the emergence of under-represented areas, highlighted by the two bounded regions in Fig. 6-middle (Exercise and Move-Turn, which differ significantly from other motion categories). These coverage gaps also impact motion annotation negatively. Another major limitation of Carrara et al. (8) is its reliance on expert sampling, which requires exhaustive manual examination of all data beforehand. This approach is very time-consuming and highly dependent on the skills and experiences of the expert.

### 8.4 IMPACT OF FRAME RATE

In most motion recognition tasks, researchers will reduce the frame rate of motion data to reduce motion blur or accelerate computation. We attempted to use this method in our research.

We conducted experiments on the HDM05-15 dataset using the same experimental settings as (Ours+$\mathcal{R}$+Conv), but we pre reduced the frame rate of the motion data to 24 fps during training. Compared to (Ours+$\mathcal{R}$+Conv, micro-F1 = 84.37%), while change to 24 fps, the micro-F1 is only 77.83%

While reducing the frame rate by 24 fps is practical for most motion recognition tasks, motion annotation requires annotating every frame, which demands richer temporal information. Reducing the frame rate may disrupt potential temporal information.

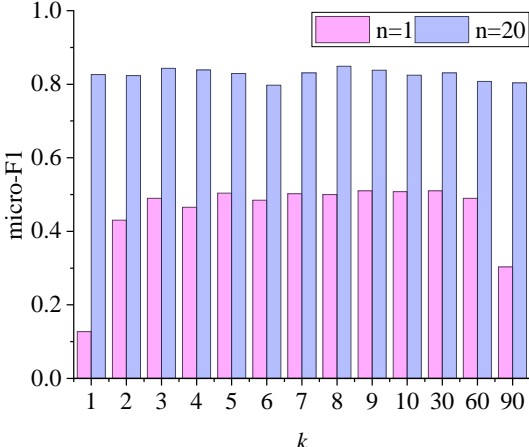

Figure 7: Annotation performance under different cluster Number settings on HDM05-15 dataset. n is the number of annotated motion sequence

### 8.5 IMPACT OF CLUSTER NUMBER ON RANKING INITIALIZATION

The number of clusters $k$ in k-means is a hyperparameter that affects the initialization results. As shown in Fig. 7, let $n$ be the number of annotated motion sequences, the micro-F1 scores of our method are insensitive to the choices of $k$ between 3 and 60. Thus, without loss of generality, we use $k$=3 in all our experiments.

### 8.6 PSEUDOCODE OF REPRESENTATIVENESS RANKING

---

**Algorithm 1:** $R^3$: Representation-based Representative Ranking

---

**Data:** Motion representation subspace $\mathcal{R}^*$, motion dataset $D = [x_1, x_2, ..., x_T]$ and $\hat{D} = []$, binary classifier $C$.

**Result:** Ranked motion dataset $\hat{D}$.

1 Separate $D$ into k clusters $\{D^{(1)}, ..., D^{(k)}\}$ via k-means clustering;

2 $i \leftarrow \text{argmax}_i |D^{(i)}|$;

3 $\overline{D}_i \leftarrow$ cluster center of $D^{(i)}$ in $\mathcal{R}^*$;

4 $a \leftarrow \text{argmin}_a ||x_a - \overline{D}_i||_2^2$ in $\mathcal{R}^*$;

5 $\hat{D} \leftarrow \hat{D}.\text{append}(x_a)$, $D \leftarrow D.\text{remove}(x_a)$;

6 **while** $D \neq []$ **do**

7     Train $C$ to distinguish between elements of $\hat{D}$ and $D$ in $\mathcal{R}^*$;

8     $x_a \leftarrow$ the median of top representative samples $x_i \in D$ w.r.t. $C$;

9     $\hat{D} \leftarrow \hat{D}.\text{append}(x_a)$, $D \leftarrow D.\text{remove}(x_a)$;

10 **end**

---

