# OpenReview forum: "Motion-R$^{3}$: Fast and Accurate Motion Annotation via Representation-based Representativeness Ranking"
_ICLR.cc/2026/Conference — ICLR 2026 Conference Withdrawn Submission_

### Official Review · Reviewer_24in · 2025-10-24

**Soundness:** 3
**Presentation:** 1
**Contribution:** 2
**Rating:** 2
**Confidence:** 5

**Summary:**

This paper introduces Motion-R3, a data-centric framework designed to improve the efficiency of motion capture annotation. The core methodology involves two main stages: first, an unsupervised, dual-level contrastive learning approach is used to learn an informative representation space for motion data. Second, a method termed "Representation-based Representativeness Ranking" (R3) is employed to select a small, highly representative subset of the data. This subset is manually annotated and used to train a simple classifier, which then automatically annotates the remaining unlabeled data. The authors demonstrate that their approach reduces manual annotation effort while maintaining high accuracy on datasets like HDM05.

**Strengths:**

The problem of reducing annotation effort for motion capture data is highly relevant, and the data-centric philosophy you have adopted is a valuable perspective. The proposed framework, Motion-R3, presents a pragmatic and potentially effective workflow.

**Weaknesses:**

I believe the paper, in its current form, has several significant issues that need to be addressed to strengthen its contribution and clarify its positioning within the broader machine learning literature. My comments below are intended to provide constructive feedback to help improve the manuscript.

Major Concerns:

1. Positioning of the Work in Relation to Active and Semi-Supervised Learning:
My primary concern is with the framing of the paper. The proposed pipeline appears to be a well-structured combination of two established machine learning paradigms:
The selection of a small, representative data subset for an "expert oracle" to label is the central goal of Active Learning (AL).
Using a model trained on this small labeled set to automatically generate labels for a much larger unlabeled set is a classic Semi-Supervised Learning (SSL) technique, often referred to as self-training or pseudo-labeling.
However, the manuscript avoids this established terminology and does not include a discussion of the rich literature in these fields in the "Related Work" section. This omission leads to two significant problems:
(a) Inflated Novelty: For readers unfamiliar with AL and SSL, the paper might give an impression of greater framework-level novelty than is perhaps warranted. The core workflow itself is not entirely new, but rather an application and integration of these ideas to the motion domain.
(b) Unclear Positioning: More importantly, by not engaging with this literature, the paper fails to properly contextualize its contribution. It is unclear how Motion-R3 builds upon or differs from prior work in AL and SSL. For example, how does your "representativeness ranking" compare to classic AL query strategies like uncertainty sampling, diversity sampling (e.g., core-set), or representative-based sampling? How does your one-pass annotation model compare to established SSL methods?
Recommendation: I strongly recommend reframing the introduction and related work to explicitly acknowledge these connections. This will not diminish your contribution; rather, it will properly position it, clarify what specific components are novel (e.g., your dual-level representation learning, your specific R3 ranking heuristics), and demonstrate a deeper engagement with the foundational literature.
2. Insufficient Justification for Key Design Choices:
The methodology section describes what was done in detail, but often lacks sufficient explanation of why specific choices were made. A strong methods section should not only be a procedural report but should also convey the authors' insights and reasoning.
(a) Choice of Representation Learning Framework: The paper states MoCo was chosen because it is a "latest contrastive learning approach." This is not a strong justification. MoCo was proposed in 2020 and, from a 2025 perspective, is a foundational but not a "latest" method. There are known discussions about its mechanics, such as the buffer update strategy. Why was MoCo specifically suitable for this task compared to other contrastive methods (e.g., SimCLR, BYOL, DINO)? What properties of MoCo were advantageous for learning motion representations?
(b) Rationale for "Representativeness": The link between Farthest Point Sampling (FPS) and "representativeness" is asserted but not deeply motivated. Why should samples that are far apart in this specific, contrastively learned space be considered the most representative for a classification task? An intuitive or theoretical argument here would be very helpful.
(c) Design of Contrastive Learning Strategy: The paper details the data augmentations and the dual-level loss, but the rationale behind these specific designs could be elaborated. What insights about motion data led to these specific augmentation techniques (e.g., perturbation, downsampling, reverse) and the combination of sequence-level and frame-level losses?
Recommendation: The methodology section would be significantly strengthened if you could provide a more in-depth discussion of your design choices. I would like to better understand the authors' perspective and reasoning that led to this specific instantiation of the framework.
3. Limited and Dated Datasets:
The experimental evaluation relies heavily on the HDM05 dataset, which is relatively old and has known limitations. While the inclusion of the BABEL dataset is a good step, the claims of generalizability would be much more convincing with a more thorough evaluation on more diverse and recent human motion datasets.

Minor Concerns:

1. Scope of "Motion": The term "motion" is used broadly in the title and text. This could create ambiguity, as it might be interpreted to include general object motion prediction (e.g., for vehicles, fluids). I suggest specifying the scope to "human motion classification" early and consistently (e.g., in the title and abstract) to set clear expectations for the reader.
Single-Pass Annotation: The semi-supervised annotation process appears to be a single-pass workflow. Many modern SSL methods involve iterative refinement, where pseudo-labels are re-evaluated, filtered, and used to retrain the model. Was an iterative approach considered? A brief discussion on why a single-pass approach was chosen (e.g., for efficiency, or because initial experiments showed no benefit) would add completeness to the experimental design.

**Questions:**

See weaknesses.

---

### Official Review · Reviewer_ufSR · 2025-10-24

**Soundness:** 2
**Presentation:** 2
**Contribution:** 2
**Rating:** 2
**Confidence:** 3

**Summary:**

This paper addresses the challenge of selecting the most representative data to annotate for the action recognition task. To get better feature representations, the author proposes using two contrastive losses to train the model in a supervised way. The author then uses a K-means-based method to select the most representative samples based on their features. Experiments are conducted to demonstrate the effectiveness.

**Strengths:**

The paper aims to select the most representative samples from a large pool of unlabeled data for labeling. This is a clear and practical motivation, and it is a promising research field.

**Weaknesses:**

- The term "motion annotation model" is misleading. The model is more accurately described as a self-supervised, pre-trained model designed to extract high-quality feature representations. Consequently, the study's primary contribution appears to be in data-efficient learning, specifically selecting the most informative unlabeled samples for annotation to train a model. However, the paper lacks necessary discussion and comparison with related work in self-supervised learning. Critically, it also omits a key baseline: directly using features extracted from existing fully-supervised action recognition models. Such models are supposed to produce strong features since trained on labeled data.
- In Section 3.2.1, the author introduces their dilated feature encoder by "borrowing the idea of dilated convolution." However, the author proposes to enhance the current frame with adjacent frames, whereas dilated convolution is a conv operation. There is a conceptual gap between these two applications, and simply stating they "borrow the idea" is insufficient to explain the motivation. Consequently, the paper lacks an explanation for why the proposed approach should work. In addition, the authors choose to enhance the current frame with raw temporal point clouds in addition to temporal convolution or attention (which enhance the current frame in the feature dimension). The necessity of this approach is also not discussed.
- As shown in Eq.1, a 3D point cloud frame is converted into a 4D representation. Subsequently, a sequence of these frames is formed into a 5D tensor of shape $\mathbb{R}^{T \times 2nl \times 3}$. However, there is no explanation about how this 5D tensor is processed by the spatial-temporal transformer, which is expected to process a 4D input of shape $\mathbb{R}^{T \times 3}$.
- The explanation of Eq.2 is not clear. For example, the author mentions that "$t_{pd} \cdot t_{md}$ means that ${s_i}$ is perturbed with missing data perturbation". How can a product of probabilities represent a data perturbation operation? From Eq.2, I believe the author means that when $p_i < t_{pd} \cdot t_{md}$, the data missing perturbation is applied. If so, using the empty set symbol would be much clearer than using 0.
- The paper contains some inconsistent notations, which makes parts of it difficult to follow. For example, Eq.1 defines s’{i,j} as the j-th frame of the i-th sequence x_i. However, Eq.2 introduces s’i, which is also described as a frame. The difference between s’{i,j} and s’_i is not explained. Clarifying this distinction would improve the paper's overall clarity.
- According to line 281, all augmented sequences are processed with both perturbation and downsampling. An ablation study is necessary to show the individual effectiveness of each data augmentation
- For the sequence-level contrastive loss, the batch consists of only three samples: two positive augmentations and one negative augmentation, all derived from the same original sample. This design differs significantly from mainstream contrastive methods like MoCo and SimCLR: (1) Batch size: Mainstream methods like MoCo benefit from a very large batch or memory bank, whereas this method uses a minimal set of three samples; (2) Negative sample: Typically, negative samples are different instances (e.g., other sequences / labels in the batch). Here, the negative is an augmentation of the same instance as the positives. This leads to a critical question: What is the intended purpose of this contrastive loss? It is not clear what specific capability the model is meant to learn from contrasting two views of a sample against a third, differently augmented view of that same sample. And as the author proposes two major changes to the mainstream approach to contrastive loss, essential discussion and experiments are needed to explain why these changes were made and how they work.
- The proposed "frame-level contrastive loss" is misleading. Since it only encourages feature similarity between neighboring frames and does not use negative samples, it is not technically a contrastive loss.

**Questions:**

- In Eq.4, $n_{ds}$ is set to a constant value of 512. However, since the offset and downsampling interval can vary, how can we ensure that more than 512 frames remain after downsampling?
- How is the binary classifier C trained?
- The ranking method identifies the densest cluster center $D_m$ and its closest samples, but their role in the final ranking is unclear

---

### Official Review · Reviewer_yvGp · 2025-10-29

**Soundness:** 3
**Presentation:** 3
**Contribution:** 2
**Rating:** 4
**Confidence:** 3

**Summary:**

The paper presents a way to choose examples to be labeled in the context of action recognition. First an embedding space is learned using contrastive learning and then examples are chosen to be far from each other in this embedding space (using an algorithm similar to farthest point). The experiments in the main paper are focused on three variants of the same dataset (HDM05) and show that one can reach 85% F1 with only 20% of the data.

**Strengths:**

I found the paper well-written and easy to understand. I liked figure 1 which gives a nice overview of the method. The idea of doing representativeness on a latent representation rather than the raw motion vectors is a sensible idea.

The experimental results show a nice improvement compared to training on the full dataset and there are several ablation studies trying to elucidate the importance of design choices made in the algorithm.

**Weaknesses:**

My main concern is significance. Although figure 1 states that the annotator only needs to do 20% of the work, this is for the choice of 0.85  F1 score. As figure 3 shows, more annotations do improve performance and the curve is far from saturating at 20%. Furthermore, using a   random selection strategy already allows one to achieve 0.85 F1 with only 40% of the examples. If we were to use a large F1 score, then the advantage of proposed method over random selection would be even smaller.

As the authors point out, "farthest point" type algorithms may end up selecting outliers. For this dataset, the authors managed to avoid this using K-means based initialization but there are no theoretical guarantees that this will work for other datasets and I can imagine settings where the K-means will also end up selecting outliers when K is set incorrectly.

**Questions:**

Can you add to figure 3 horizontal lines corresponding to 0.9 and 0.95 F1 score?

Can you clarify the experiment that is summarized in table 1? What do you mean by "on the HDM05 dataset under the same conditions."
In figure 3 you show F1 score as a function of labeled examples but in table 1 there is a single number for each method.

---

### Official Review · Reviewer_7B5c · 2025-11-01

**Soundness:** 2
**Presentation:** 3
**Contribution:** 2
**Rating:** 4
**Confidence:** 3

**Summary:**

This paper presents a data-centric motion annotation method named Representation-based Representativeness Ranking (R^3), which prioritizes motion samples according to their significance within a learned representation space R. To enhance the informativeness of this space, the authors introduce a novel dual-level contrastive learning framework that operates at both the sequence and frame levels. Owing to its high efficiency, the proposed Motion-R^3 method can readily adapt to frequently changing requirements, thereby facilitating agile development in motion model annotation.

**Strengths:**

This paper presents an effective and efficient method for motion annotation. It is well-structured and clearly written, making the technical content accessible. I particularly appreciate the figures, which are not only clear and informative but also visually compelling, elegantly illustrating the core concepts of the proposed approach.

**Weaknesses:**

* The methodological innovation of this work appears limited. The self-supervised learning framework directly adopts the established MoCo architecture, while the dual-level (sequence and frame) contrastive learning paradigm has been extensively explored in video representation learning. The adaptation of these components to the motion domain, while valuable for the task, does not constitute a significant conceptual advancement. A more detailed discussion on how the proposed approach differs from and advances beyond these existing contrastive learning strategies would be necessary to better highlight its novelty.
* A central claim of this work is the improved efficiency of the annotation pipeline. However, the experimental validation is conducted on a relatively limited scale of data, which undermines the generalizability and practical impact of this claim. To strengthen the contribution, it is crucial to demonstrate the method's effectiveness on larger and more diverse datasets. Furthermore, in the context of modern data-centric approaches, a key question arises: how does the proposed method compare to, or complement, the use of powerful pre-trained Vision-Language Models (VLMs, e.g., Gemini, GPT-4V) for automated annotation or data screening? An ablation or discussion addressing this point would significantly solidify the paper's position.
* I seek clarification on the core concept of "motion annotation" as defined in this paper. My understanding is that it is equated with assigning a motion category label. If this is accurate, the term "annotation" might be too narrow, as it typically encompasses a broader set of labels in the motion community, including at the very least detailed human pose keypoints. This potential discrepancy between the broad claim of the title and the specific task of category labeling could be perceived as an overclaim. Would it be more appropriate by precisely defining the scope of "motion annotation" by consider adjusting the title to more accurately reflect the specific contribution (e.g., focusing on "category-efficient learning" or "representative sampling for classification").

**Questions:**

Please refer to the weakness.

---

### Note · Authors · 2025-11-12

I have read and agree with the venue's withdrawal policy on behalf of myself and my co-authors.